# Protein Inverse Folding From Structure Feedback

**Junde XU**[1,2]    **Zijun Gao**[1]    **Xinyi Zhou**[1]    **Jie Hu**[3]    **Xingyi Cheng** [4]
**Le Song**[4]    **Guangyong Chen**[2]    **Pheng-Ann Heng**[1]    **Jiezhong Qiu**[2*]
[1] CUHK    [2]Hangzhou Institute of Medicine, CAS    [3] Zhejiang Lab    [4] MBZUAI
qiujiezhong@him.cas.cn

## Abstract

The inverse folding problem, aiming to design amino acid sequences that fold into desired three-dimensional structures, is pivotal for various biotechnological applications. Here, we introduce a novel approach leveraging Direct Preference Optimization (DPO) to fine-tune an inverse folding model using feedback from a protein folding model. Given a target protein structure, we begin by sampling candidate sequences from the inverse-folding model, then predict the three-dimensional structure of each sequence with the folding model to generate pairwise structural-preference labels. These labels are used to fine-tune the inverse-folding model under the DPO objective. Our results on the CATH 4.2 test set demonstrate that DPO fine-tuning not only improves sequence recovery of baseline models but also leads to a significant improvement in average TM-Score from 0.77 to 0.81, indicating enhanced structure similarity. Furthermore, iterative application of our DPO-based method on challenging protein structures yields substantial gains, with an average TM-Score increase of 79.5% with regard to the baseline model. This work establishes a promising direction for enhancing protein sequence design ability from structure feedback by effectively utilizing preference optimization[†].

## 1   Introduction

Just as language serves as the foundation of human communication and social organization, proteins constitute the fundamental molecular machinery governing life's essential processes across all biological systems. The emerging task of protein sequence design (inverse folding) aims to find amino acid sequences that reliably fold into target three-dimensional architectures while exhibiting predetermined functional capabilities. This paradigm has catalyzed transformative applications spanning next-generation therapeutics development [31] to the creation of engineered biocatalysts revolutionizing industrial processes [43].

Recent advances in protein inverse folding have shifted from physical methods to deep-learning-based methods [3]. Traditional physics-based methods, such as those implemented in Rosetta [41], rely on energy-based modeling to identify sequences compatible with a given structure. In contrast, deep learning-based models have demonstrated significant progress by leveraging geometric structure encoders and massive sequence datasets. Methods like ProteinMPNN [10], ESM-IF [17], and LigandMPNN [11] employ SE(3)-equivariant networks [9] to capture the spatial properties of protein backbones, enabling accurate and efficient sequence design. More recently, the availability of large-scale protein sequence databases has spurred the development of protein language models such as ESM-3 [16], ProGen2 [30], which learn rich sequence representations and show promise in generating structurally compatible and functionally diverse protein sequences [36, 39].

---

[*]Corresponding author.
[†]Code available at https://github.com/Eikor/iplm-rl

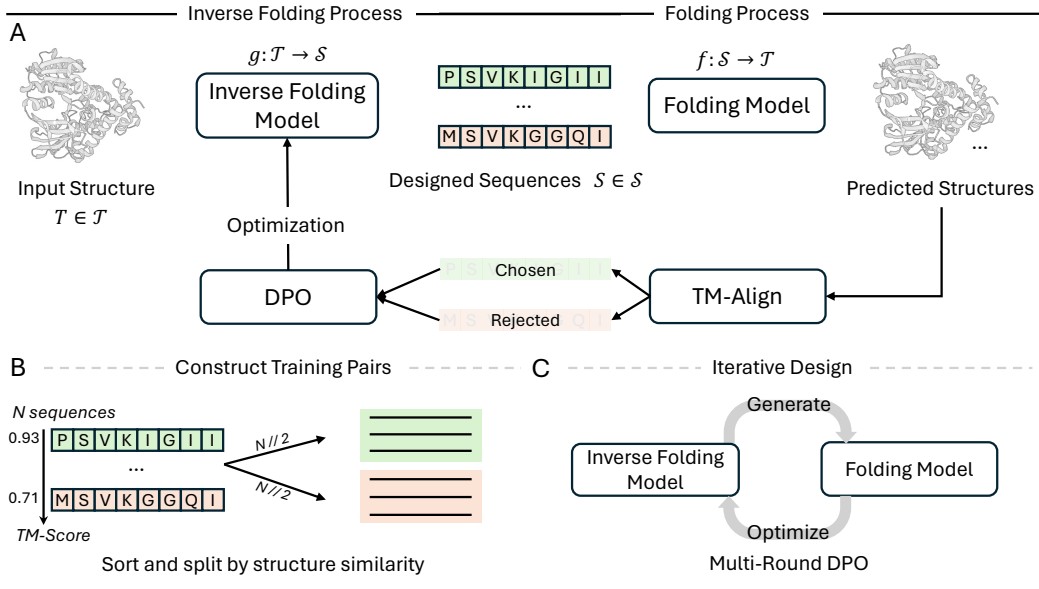

Figure 1: **Overview of our methods**. **A.** We connect the inverse-folding process and folding process with DPO, utilizing the structure similarity (TM-Scores) predicted by the folding model to guide the optimization of the inverse-folding model. **B.** We classify generated sequences into chosen and rejected based on their TM-Scores. **C.** Multi-round DPO for iterative refinement of designed sequences.

Current inverse folding models are typically formalized as generative models, which are trained to maximize the conditional probability of protein sequences given their corresponding structures. The underlying assumption is that proteins with similar sequences will also have similar structures. This allows for the design of new proteins that fold into desired structures by sampling from the learned generative model. However, previous studies [46, 22] have identified proteins with similar sequences but significant differences in structure and function. This suggests that sequence similarity (which is the learning objective of many inverse folding models) is not always a reliable indicator of structural similarity. Moreover, generative models for sequences — whether for protein sequences or text — may experience hallucinations and produce low-quality or repetitive responses. They can also sometimes fail to follow instructions accurately [27, 5, 51, 49]. These challenges underscore the need for further improvements to enhance the reliability and accuracy of these models. Due to these challenges, practical protein design workflows [38, 40] often involve generating a large number of sequences, followed by a sophisticated selection process to prioritize the most promising candidates. Only then can costly experimental validation be conducted. While necessary, this approach introduces additional steps and complexity into the protein design process, potentially limiting the broader application of inverse folding algorithms.

Fortunately, recent advances in preference-based learning, represented by Direct Preference Optimization (DPO) [37], offer an opportunity to mitigate those problems. These methods train models on a pre-generated dataset, capture the preference information by constructing chosen and rejected training pairs, and teach the model to classify good or bad responses. Some efforts have been made to apply DPO on protein language models [28, 44] and even inverse folding models [47, 35]. However, these existing efforts primarily rely on datasets derived from wet-lab experiments, where the function or structure preferences are manually curated through costly biochemical assays. As a result, the application of DPO and other preference-based training strategies has been restricted to small-scale datasets, limiting the scope and generalizability of the resulting models. In contrast, recent breakthroughs in protein folding methods, such as AlphaFold [21], ESMFold [25], and RoseTTAFold [23] have enabled fast, accurate, and *in silico* assessment of sequence-structure compatibility at scale.

Motivated by this, we explore a fully *in silico* training scheme that uses folding model feedback to enhance the sequence design capabilities of inverse folding models. Specifically, we utilize an inverse folding model to generate a diverse set of protein sequences. These sequences are then fed

into a folding model to predict their corresponding structures. We evaluate these structures w.r.t the ground-truth structure using TM-Align [52] and classify the sequences into 'chosen' and 'rejected' based on the TM-Score. Finally, we employ Direct Preference Optimization (DPO) to fine-tune the inverse folding model. Building on this, we extended our investigation to explore the effects of multi-round DPO. Through extensive experiments, we demonstrate the effectiveness of using folding model feedback within a DPO framework to improve inverse protein folding. Our DPO models consistently achieve higher sequence recovery across all 3 datasets (CATH 4.2 [32], TS50 and TS500 [12]) compared to their baselines. Notably, a detailed study on the CATH 4.2 dataset shows a steady improvement in structure similarity, suggesting that our method learns to prioritize backbone and fold-related features. Furthermore, scaling experiments reveal that the quality of contrastive samples is more crucial than sheer quantity for achieving high structural fidelity. Finally, experiments of multi-round DPO showcase substantial gains in TM-Score on challenging protein structures, evidencing the iterative refinement ability of our methods.

## 2 Preliminaries

**Protein Folding and Protein Inverse Folding.** The protein folding problem seeks to predict a protein's 3D structure given its sequence. The goal is to learn a mapping $f : \mathcal{S} \to \mathcal{T}$, where $\mathcal{S}$ and $\mathcal{T}$ are the spaces of protein sequences and structures, respectively. Inverse folding (aka "protein sequence design") predicts a protein's sequence given its structure, which learns an inverse mapping $g : \mathcal{T} \to \mathcal{S}$ from structure space $\mathcal{T}$ to sequence space $\mathcal{S}$. In particular, denote $\mathcal{D} \subset \mathcal{T} \times \mathcal{S}$ to be the dataset of known structure–sequence pairs (e.g., from PDB or CATH). The inverse folding model (parameterized by $\theta$) is trained to maximize the conditional likelihood as follows:

$$\max_{\theta} \; \mathbb{E}_{(T,S) \sim \mathcal{D}} \left[ \log P_\theta \left( s_1, \ldots, s_L \mid T \right) \right], \tag{1}$$

where $T \in \mathcal{T}$ denote the protein structure, and $S = \{s_1, s_2, \ldots, s_L\}$ denote the sequence of length $L$. A common choice for factorizing the sequence probability is to adopt an autoregressive approach, leading to what is often referred to as the language model loss:

$$\mathcal{L}_{\text{inv}} = -\frac{1}{L} \sum_{i=1}^{L} \log P_\theta \left( s_i \mid T, \; s_{<i} \right), \tag{2}$$

which encourages the model to recover ground truth residues, conditioned on both the target structure and previously predicted residues.

**Reinforcement Learning from Human Feedback.** Let $\pi_\theta$ be the language model parameterized by $\theta$. The goal of Reinforcement Learning from Human Feedback (RLHF) [8, 33] is to fine-tune large language models (LLMs) to better align with human preferences. Typically, given the prompt $x$, the language model $\pi_\theta(y|x)$ generates a response $y$, and human feedback in the form of pairwise preferences between completions is used to train a reward function $r_\phi(x, y)$. This reward model helps the LLM prioritize preferred completions $y_w$ over less preferred ones $y_l$, with the probability of preference modeled by:

$$p(y_w > y_l | x) = \sigma(r_\phi(x, y_w) - r_\phi(x, y_l)) \tag{3}$$

where $\sigma$ is the sigmoid function defined as $\sigma(z) = \frac{1}{1+e^{-z}}$. The objective is to maximize this probability, and the reward function $r_\phi$ is learned by minimizing the negative log-likelihood over the preference dataset $\mathcal{D}_{pair}$:

$$L_R(r_\phi, \mathcal{D}_{pair}) = -\mathbb{E}_{(x,y_w,y_l) \sim \mathcal{D}_{pair}}[\log \sigma(r_\phi(x, y_w) - r_\phi(x, y_l)] \tag{4}$$

To prevent the fine-tuned model $\pi_\theta$ from deviating too much from the initial supervised model $\pi_{\text{ref}}$, a KL-divergence constraint is added, preserving linguistic quality while aligning with human preferences. However, RLHF can be complex and unstable. A more recent approach, Direct Preference Optimization (DPO) [37], simplifies the process by directly optimizing for human preferences without needing a reward model or reinforcement learning [42], offering a more streamlined alternative. The DPO loss directly minimizes the negative log-likelihood of the preference model:

$$L_{\text{DPO}}(\pi_\theta; \pi_{\text{ref}}) = \mathbb{E}_{(x,y_w,y_l) \sim \mathcal{D}_{pair}} \left[ -\log \sigma \left( \beta \log \frac{\pi_\theta(y_w \mid x)}{\pi_{\text{ref}}(y_w \mid x)} - \beta \log \frac{\pi_\theta(y_l \mid x)}{\pi_{\text{ref}}(y_l \mid x)} \right) \right] \tag{5}$$

Here, $\beta$ is a parameter that determines the level of information retention from the reference model, helping to balance novelty and adherence to the reference behavior. This setup allows for a more straightforward optimization process that is inherently more stable.

## 3 Methods

**Constructing Preferences Datasets.** Given a structure $T$ drawn from dataset $D$, we first apply the inverse-folding model $\pi_\theta$ to sample $N$ initial predictions,

$$\{S_n\}_{n=1}^N \sim \prod_{n=1}^N \pi_\theta(\,\cdot\,|T) \tag{6}$$

Next, structures corresponding to the sampled sequence $S_n$ are predicted using a folding model, $T_n = g(S_n)$, where $g$ is a protein folding model (e.g., ESMfold). The TM-Score $c_n \in [0, 1]$ was then calculated with TMAlign [52] for each predicted structure in relation to its corresponding ground-truth structure $c_n = \text{TM-Score}(T_n, T)$. To create balanced training pairs, the top-ranked $50\%$ of sequences are labeled as chosen, while the bottom $50\%$ are labeled as rejected (Fig. 1). Formally, for $N$ generated sequences with TM-Scores $\{(S_n, c_n)\}_{n=1}^N$, we sort the sequences by the TM-Score in a descending order and split into chosen $S^w$ and rejected $S^l$ as follows:

$$k = \text{permute indices so that } c_{k(1)} \geq c_{k(1)} \geq \cdots \geq c_{k(N)}. \tag{7}$$

Then for $j = 1, \ldots, \frac{N}{2}$ we have

$$(T, S_j^w, S_j^l) = \left(T, S_{k(j)}, S_{k(j+\frac{N}{2})}\right) \tag{8}$$

Finally, our new preference dataset can be written as $\mathcal{D}_{pair} = \{(T, S_j^w, S_j^l)\}_{j=1}^{N/2}$. This equal split ensures a balanced dataset for training.

**Finetuning with DPO and SFT Loss.** Unlike traditional NLP datasets, designed protein sequences have a much smaller vocabulary and can share similar residues even when the TM-Score is low. As the original DPO loss will minimize the probability of rejected responses, the probability of similar parts will also be minimized, eventually causing model degeneration (similar to the alignment of mathematical problems in NLP tasks, where the responses are similar despite the answer being wrong [34]). A simple method to tackle this problem is to add an SFT loss as a regularization term. Given the constructed preference dataset $\mathcal{D}_{pair}$, the DPO loss on the inverse folding model $\pi_\theta$ is:

$$L_{\text{DPO}}(\pi_\theta; \pi_{\text{ref}}) = \mathbb{E}_{(T,S^w,S^l) \sim \mathcal{D}_{pair}} \left[ -\log \sigma \left( \beta \log \frac{\pi_\theta(S^w \mid T)}{\pi_{\text{ref}}(S^w \mid T)} - \beta \log \frac{\pi_\theta(S^l \mid T)}{\pi_{\text{ref}}(S^l \mid T)} \right) \right] \tag{9}$$

The final loss can be written as:

$$L(\pi_\theta; \pi_{\text{ref}}) = \lambda \cdot \mathbb{E}_{(T,S^w,S^l) \sim \mathcal{D}_{pair}} [-\log(\pi_\theta(S_w \mid T))] + L_{\text{DPO}}, \tag{10}$$

where the $\lambda$ is a super-parameter to control the contribution of rejected responses during training.

**Iterative Training.** To investigate whether our methods can consistently improve the performance in a multi-round setting, we also employ an interactive training framework where the inverse folding models are successively refined using preference data generated by their predecessors. For each iteration $t$, the model $\pi_\theta^t$ will first generate and construct preference data $D_{pair}^t$ as described in Sec. 3, the model is then updated to $\pi_\theta^{t+1}$ with the DPO and SFT loss (Eq. 10) on generated dataset $D_{pair}^t$. Crucially, the reference model $\pi_{\text{ref}}$ is reinitialized with the weight of $\pi_\theta^t$ at the start of each iteration, ensuring consistency in preference alignment while avoiding catastrophic forgetting.

## 4 Experiments

### 4.1 Benchmark Performance

To assess the broad applicability of our method, we apply it to two representative protein design models, ProteinMPNN [10] and InstructPLM [36], containing around 1 million and 7 billion parameters. This setting helps verify the performance of our method across both large and compact model architectures. Specifically, we first generate sequences and train models on the CATH 4.2 training set and investigate the performance on the test set. To further probe generalization, we also evaluate on two additional benchmarks, TS50 and TS500 [12], which comprise 50 and 470 diverse proteins, and are often employed as additional benchmarks to further test generalization capability[53, 14, 15]

Table 1: **Sequence design performance.** We report the perplexity and recovery rate on the CATH 4.2, TS50, and TS500 datasets. The performance of our methods is shown in **bold**, while baselines without fine-tuning are indicated with an underline.

| Model | CATH 4.2 | | TS50 | | TS500 | |
|---|---|---|---|---|---|---|
| | Perplexity | Recovery | Perplexity | Recovery | Perplexity | Recovery |
| ProteinMPNN [10] | 5.41 | 40.27 | 5.10 | 44.72 | 4.57 | 46.58 |
| PiFold [15] | 4.55 | 51.66 | 3.86 | 58.72 | 3.44 | 60.42 |
| LM-Design [53] | 4.52 | 55.65 | 3.50 | 57.89 | 3.19 | 67.78 |
| KW-Design [13] | 3.46 | 60.77 | 3.10 | 62.79 | 2.86 | 69.19 |
| InstructPLM [36] | 2.63 | 53.58 | 2.46 | 60.20 | 2.14 | 66.13 |
| **ProteinMPNN-DPO** | **5.09** | **41.29** | **4.85** | **45.91** | **4.26** | **48.23** |
| **InstructPLM-DPO** | **2.71** | **55.21** | **2.52** | **62.01** | **2.17** | **66.37** |

Table 2: **Structure similarity of designed sequences.** We report the TM-Scores on different splits of the CATH 4.2 test set, the best results are shown in **bold**.

| Model | Short | Single-Chain | All | TM-Score$< 0.5$ | TM-Score$> 0.5$ |
|---|---|---|---|---|---|
| InstructPLM | 0.54 | 0.56 | 0.77 | 0.37 | 0.87 |
| SFT | 0.54 | 0.57 | 0.78 | 0.39 | 0.87 |
| DPO $\lambda = 10$ | 0.57 | 0.61 | 0.80 | 0.43 | 0.88 |
| $\lambda = 1$ | **0.58** | **0.63** | **0.81** | **0.45** | **0.89** |
| $\lambda = 0$ | 0.57 | 0.62 | 0.81 | 0.44 | 0.88 |

beyond the CATH dataset. Details of hyperparameters are shown in Tab. 7. An interesting observation is that ProteinMPNN fails to converge on self-generated datasets (Sec. 3). We infer that the learning capacity of ProteinMPNN is too small to learn from its own predictions. To tackle this problem, we construct a simpler task, rather than fine-tune ProteinMPNN only on model predictions, we also add wild-type sequences as additional chosen samples, details are provided in the Appendix A.4.

As Tab. 1 shows, both DPO models yield consistent gains over their respective baselines. Specifically, InstructPLM-DPO raises recovery by 3.0% (from 53.58% to 55.21% ), and similar gains are also achieved in ProteinMPNN-DPO (from 40.27% to 41.29%). In terms of perplexity, it is no surprise that InstructPLM-DPO has a slight increase, as the model is trained exclusively on self-generated sequences, which inevitably cause a distribution shift from the reference model. However, despite the distribution shift, the perplexity remains within competitive ranges. On the other hand, as ProteinMPNN has an extra supervision of wild-type sequences, the perplexity of ProteinMPNN-DPO successfully decreased. On TS50, InstructPLM-DPO increases recovery from 60.20 % to 62.01 %, and on TS500 from 66.13 % to 66.37 %, while perplexity rises only 0.06 and 0.03, respectively. ProteinMPNN-DPO also exhibits similar performance improvement, recovery rate increases from 44.72 to 45.91 on TS50, and even more steady increases from 46.58 to 48.23 on the TS500 dataset. This confirms that preference optimization enhances design quality across both large and compact model architectures.

## 4.2 DPO Achieves Higher Structure Similarity

To evaluate whether our methods acquire transferable, structure-related features solely from self-generated training sequences, we investigate the structure similarity measured by TM-Score of InstructPLM-DPO on the CATH 4.2 test set, all structures are predicted using ESMfold. As Tab. 2 shows, despite never seeing native test-set examples during training, InstructPLM-DPO yields consistently higher TM-scores across all splits of the CATH 4.2 test set. Specifically, the overall TM-Score reaches 0.81 for our best model, which witnesses a steady performance gain compared with the baseline model (0.77). Moreover, for different splits of the test set, our methods improve the baseline from 0.54 to 0.58 on the Short split, which contains small proteins less than 100 amino acids, indicating a performance improvement of both short and long proteins. For Single-chain, the model achieves an improvement of $12.5\%$ compared to the baseline model (from 0.56 to 0.63). Notably, all of the performance gains are acquired from the training set, the only supervision comes from the

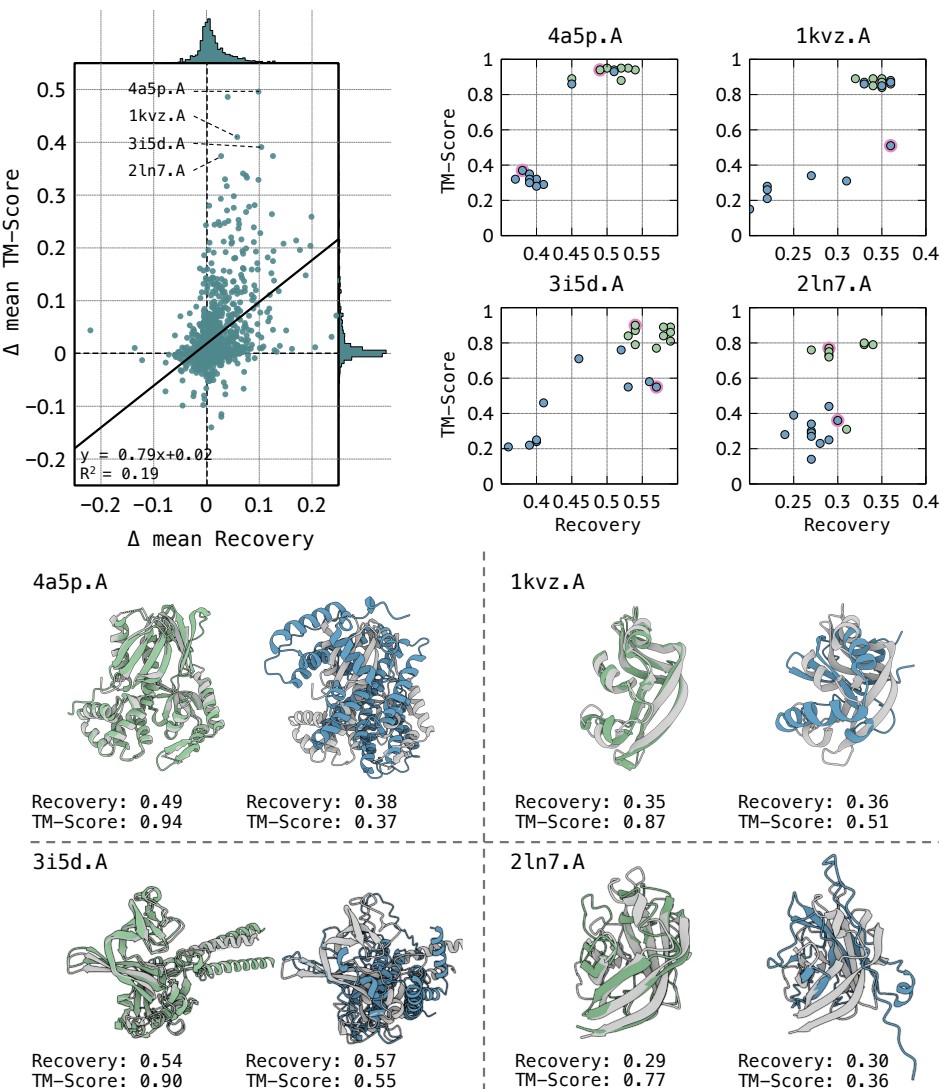

Figure 2: **Per-structure changes in TM-score vs. changes in sequence recovery on the CATH 4.2 test set. Top left:** Change in TM-Score (mean TM-Score over 10 predictions) vs. change in recovery (mean Recovery over 10 predictions) for each structure. **Top right:** TM-Score versus recovery of 10 sequences sampled from the baseline and the DPO model, samples selected for visualization are marked by circle. **Bottom:** Examples structure predicted by ESMfold, ground-truth structures are shown in gray.

folding model. These improvements demonstrate the effectiveness and generalization ability of our methods.

To better understand the role of chosen and rejected samples, we conducted an ablation study on different weights of $\lambda$, where SFT indicates that only using chosen examples to train models (Eq. 10). We also divide the test set into a low-similarity group and a high-similarity group based on the baseline performance. As Tab. 2 shows, simply training models on the chosen sample will also benefit the overall structure similarity (from 0.77 to 0.78), but is still left behind by jointly using the chosen and rejected samples simultaneously. Introducing the rejected sample in optimization leads to a significant performance gain, even under a chosen example dominant setting ($\lambda = 10$). Specifically, the rejected sample is more useful in low-similarity samples (from 0.37 to 0.45), showing that rejected samples help model lowering the probability of undesired generations. Increasing the weight of rejected samples will further improve the performance ($\lambda = 1$ vs. $\lambda = 10$). However, the performance starts to drop when the rejected samples outweigh ($\lambda = 0$) too much. These results show the importance of balancing the contribution of chosen and rejected samples during training.

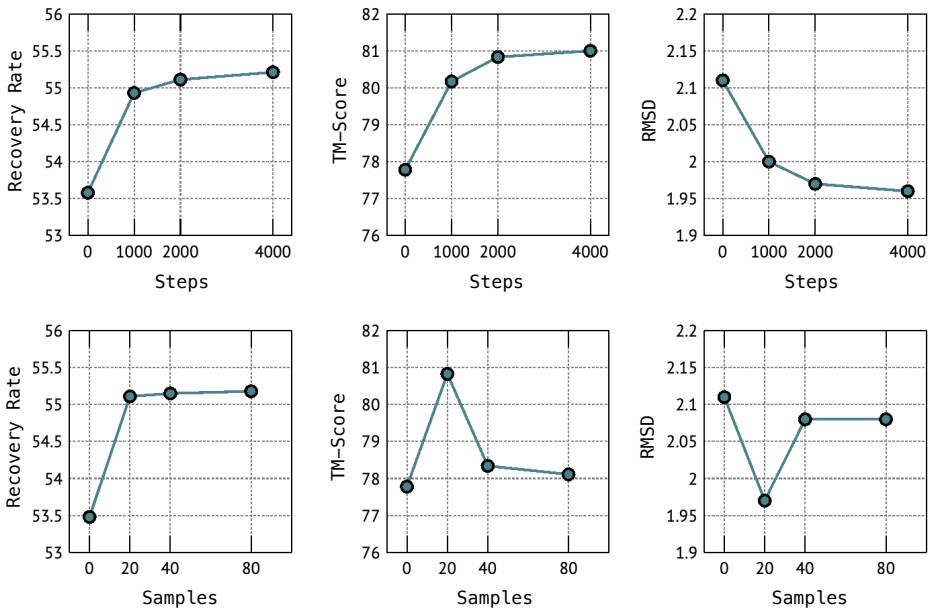

Figure 3: **Performance at different training scales.** The upper row shows protein design performance changes at different training steps, and the lower row shows protein design performance at different numbers of contrastive samples ($N$ in Eq. 6) of each structure.

## 4.3 A Closer Look at Proteins Generated by DPO Model

To evaluate the impact of our methods on the DPO model, we analyzed the relationship between the changes in structure similarity of sequences designed by InstructPLM-DPO, as measured by the TM-Score ($\Delta$ mean TM-Score), and the corresponding changes in sequence recovery ($\Delta$ mean Recovery). These changes were calculated by taking the mean of 10 independently generated sequence samples for each protein structure in the CATH 4.2 test set. In general, a linear regression analysis reveals a positive correlation ($y = 0.79x + 0.02$), suggesting that improvements in sequence recovery tend to be associated with enhanced structure similarity, which aligns with the intuitive expectation that better sequence recovery often leads to more accurate folds (top left in Fig. 2). However, the low coefficient of determination ($R^2 = 0.19$) indicates a weak association. This implies that substantial increases in TM-Score can frequently occur with relatively small changes in sequence recovery. For instance, many exhibit large TM-score boosts ($\Delta$TM-Score > 0.20) with small sequence recovery change ($\Delta$Recovery < 0.10). This is reasonable as no wild-type sequences are involved during the training phase of InstructPLM-DPO. These observations suggest that our fine-tuning process prioritizes the exploration of sequences that fold into more accurate structures, even if deviating from the wild-type sequence.

To further illustrate this observation, we visualize 4 cases with significant structure similarity improvements after fine-tuning, protein 4a5p.A, 1kvz.A, 2ln7.A, and 3i5d.A.(Fig. 2, top right and bottom.) and provide sequence alignment information for these four cases in the Appendix B.4. The top right panel of Fig. 2 shows TM-Scores and recovery rates of all 10 predictions of the baseline and DPO models of 4 cases. The 2 cases in the top row (4a5p.A and 1kvz.A) show a clear trend of more robust predictions, that InstructPLM-DPO can eliminate the low-quality predictions and strengthen high-quality predictions, which aligns with our training target. Notably, the 2 cases of the lower row (3i5d.A and 2ln7.A) evidenced the behavior that the DPO model is capable of generating more structurally accurate sequences at the same level of sequence identity. For example, the prediction of protein 3i5d.A of InstructPLM-DPO has a TM-Score of 0.90 at a recovery rate level of 0.56, while the TM-Score of the baseline model at the same level of recovery rate (0.57) only has a TM-Score of 0.55. Predictions of protein 2ln7.A also supports this trend, the fine-tuned prediction achieves a TM-Score of 0.77 with a recovery rate of 0.29, showing notable increases in TM-Score at the same recovery level.

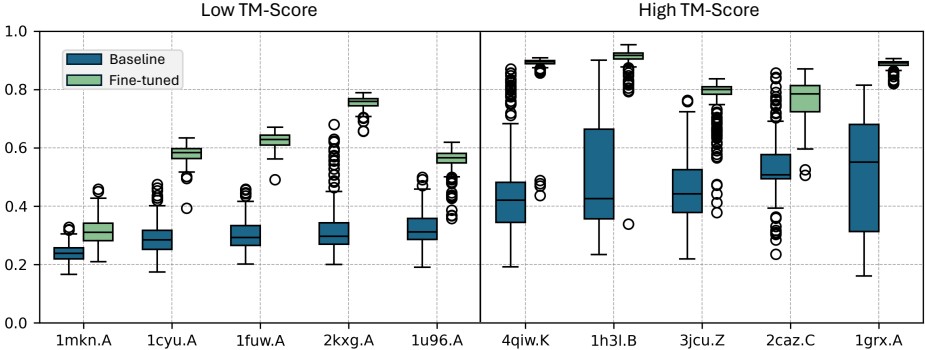

Figure 4: **Multi-round result on 10 hard structures selected from CATH 4.2 test set.** The plot shows the TM-Scores of predicted structures from the final round.

## 4.4 DPO at Scale

Scaling has been shown to improve performance in large language models and, more recently, in protein language models [2, 39, 6]. Post-training scaling techniques also yield gains in general LLMs [24], but their impact on protein design remains unclear. Here, we systematically explore two axes of scale in our framework: the number of training steps and the number of contrastive samples per structure. To disentangle these factors, we ran two sets of experiments on the CATH 4.2 training split: **Training steps,** We fixed the number of generated sequences at 20 (10 chosen vs. 10 rejected) and evaluated performance at 0, 1 000, 2 000, and 4 000 update steps. **Contrastive samples,** We fixed the training steps at 4 000 and varied the number of generated sequences (contrastive pairs) from 0 up to 80 per structure. The baseline performance corresponds to step 0 and zero contrastive samples in each scenario.

As illustrated in Fig. 3, increasing the number of training steps yields consistent improvements in both sequence recovery and structural metrics (TM-score and RMSD), showing that the training process keeps improving the performance of the model. However, improvements saturate at the training step of 4000, indicating a performance bound that we can reach in this setting. In contrast, scaling up the number of contrastive samples boosts recovery rate but degrades structural fidelity—TM-scores decline and RMSD increases. A possible explanation is that larger contrastive sets dilute the useful training signals by introducing too many low-quality samples, for example, rejected samples with high TM-Scores, thus hindering the learning process eventually. These experiments highlight that *data quality* outweighs sheer quantity. Our experiments shed light that future work should therefore focus on enhanced sample selection or multi-objective scoring strategies to maintain high-quality contrastive examples while preserving fold accuracy.

## 4.5 Iterative Protein Design

To evaluate the capacity for iterative improvement, we selected 10 difficult structures from the CATH 4.2 test set to assess the performance limits of our approach. First, we divided all structures into "Low TM-Score" and "High TM-Score" groups based on the baseline performance. Within each group, 5 structures with the lowest TM-Scores were chosen. Implementation details, including the number of rounds, generations, training steps per round, sampling temperatures, and top-p values, are detailed in Tab. 7.

The final results, presented in Fig. 4, demonstrate a significant performance gain across all 10 structures. The mean TM-Score improved by 79.5%, rising from 0.39 to 0.70. The High TM-Score group showed the most substantial improvement. Since the design model was already capable of generating high-quality sequences for these structures, DPO effectively reduced the occurrence of low-quality predictions. For instance, in designing proteins 4qiw.K and 1h3l.B, the baseline model could produce sequences with TM-Scores exceeding 0.8. However, the overall performance was diminished by a high proportion of low-quality predictions (over half with TM-Scores below 0.5). After fine-tuning, the median TM-Score for 4 out of 5 structures in this group surpassed 0.8. Importantly, the upper bound of design performance also increased, indicating that DPO not only minimizes

low-quality predictions but also more accurately aligns the space of high-quality predictions, enabling the exploration of previously unseen, superior predictions. Structures in the Low TM-Score group also exhibited substantial gains. Specifically, proteins 1cyu.A, 1fuw.A, 2kxg.A, and 1u96.A now generate sequences with TM-Scores above 0.5, compared to a baseline TM-Score of approximately 0.3. However, the DPO model still faces challenges in designing protein 1mkn.A, suggesting that a stronger baseline model may be necessary for such difficult cases. Finally, considering the limitation of computational resources, we do not perform a larger scale of experiments in multi-round DPO. Given the notable performance of iterative training, we believe this is a promising topic and leave it for future work.

## 5   Related Works

**Preference Optimization from AI Feedback.**  In response to the significant costs and limited accessibility associated with curating high-quality datasets, AI feedback has emerged as a critical component in preference fine-tuning. This approach leverages another LLM to predict the reward of a target model's response and subsequently fine-tunes the target model based on this AI-generated feedback [4, 24]. More recently, methods such as self-play [7, 48] and self-rewarding [50] have moved towards eliminating the reliance on external LLMs for feedback. For instance, SPIN [7] employs an iterative framework akin to DPO, utilizing model-generated data as rejected responses paired with human labels as the preferred responses, and iteratively refines the model. Self-rewarding [50] further abstracts away the need for chosen human labels by using the target model itself to determine preferred and dispreferred samples.

**Preference Optimization in Protein Language Models.** In contrast to the extensive exploration of preference optimization in natural language processing, protein language models (pLMs) have received comparatively less attention in this regard, primarily due to the scarcity of high-quality, labeled datasets. Nevertheless, initial efforts are being made in this direction. Widatalla et al. [47] proposed ProteinDPO to fine-tune ESM-IF for designing proteins with enhanced stability, while Mistani et al. [28] applied DPO on ProtGPT2 for binder design. Distinct from these prior studies that heavily depend on manually annotated datasets, our approach leverages synthetic data generated by the target model itself. Concurrently, Stocco et al. [44] introduced DPO_pLM to align pLMs on various desired properties, including structure similarity. However, their application of DPO focused on ZymCTRL [29], a model specifically designed for functional enzyme design, whereas our work explores a broader spectrum of protein design tasks. Similarly, Park et al. [35] explored the application of DPO on ProteinMPNN for peptide design, which operates at a smaller scale in terms of both data and model size compared to our methods.

## 6   Conclusion

**Summary.** We introduce a novel framework for optimizing inverse-folding models by the feedback of the folding model through preference optimization. Our primary finding is the significant improvement in both the sequence recovery and the structure similarity of designed protein sequences, as evidenced by extensive experiments on the CATH 4.2 dataset compared to the baseline model. Moreover, the case study shows that our methods not only preserve high-quality predictions of the baseline model while eliminating the low-quality predictions, but also generate more structurally accurate samples even at the same level as the baseline model. This indicates that the model learns to prioritize the underlying principles of protein folding.

**Social impact.** Our end-to-end, fully *in silico* optimization pipeline for protein inverse folding offers significant benefits across healthcare, industry, and academic research. Notably, our work validates the effectiveness of the optimization-from-feedback paradigm, suggesting that the pipeline is flexible and can readily incorporate alternative forms of feedback beyond structural metrics, thereby enabling the development of more powerful and versatile protein design models.

**Limitations.** While increasing the number of training steps generally leads to improved performance, simply scaling the number of contrastive samples per structure can be detrimental to structure similarity, even if it slightly boosts sequence recovery. This finding underscores a limitation of our current approach. More sophisticated preference dataset construction strategies may further improve the performance.

## Acknowledgments and Disclosure of Funding

The work described in this paper was supported in part by the Research Grants Council of the Hong Kong Special Administrative Region, China, under Project T45-401/22-N; and by the Hong Kong Innovation and Technology Fund, under Project ITS/241/21. Jiezhong Qiu was supported by NSFC 62306290.

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

# A Implementation Details

## A.1 Generation Configs for Evaluation

To better evaluate the performance of DPO models, we generate 10 samples at a temperature of 0.15 for each structure in all datasets used during evaluation. We report the mean value of 10 predictions for each metric.

## A.2 Low-Rank adaptation

To keep the knowledge in the pre-trained model, we utilized LoRA [18] during the training process of InstructPLM-DPO. The key idea of LoRA (Low-Rank Adaptation) is to enhance the adaptability and efficiency of pre-trained models by integrating trainable low-rank matrices into existing model weights, allowing for targeted modifications without retraining the entire model.For each layer in the pre-trained model, let $h^{n-1} \in \mathbb{R}^{l \times d}$ be the inputs of $n$-th layer, $l$ is the sequence length, and $d$ is the hidden dimension. $h_{qkv}^n$ be the output of QKV projection module of $n$th layer, the updated output $h_{qkv}^{n*}$ can be written as

$$h_{qkv}^{n*} = h_{qkv}^n + W_B \times W_A \times h^{n-1}, \tag{11}$$

where $W_A \in \mathbb{R}^{r \times d}$ and $W_B \in \mathbb{R}^{d \times r}$ are LoRA weights, which we initialized randomly from a Gaussian distribution with mean zero and standard deviation of 0.01. We set the rank $r$ as 16 by default, resulting in a fraction of 0.1% of trainable total parameters. The strategy helps us balance model expressiveness with computational efficiency.

## A.3 Hyperparameters

We use two different training configs for single-round training and multi-round training. The main difference lies in the generation setting and training steps. Specifically, for the construction of the single-round dataset, we generate 20 responses for each structure, with temperature at 1.0 and top $p$ at 0.9. For multi-round, in order to encourage model exploration, we set the temperature at 1.1 and top $p$ at 1 and generate 200 responses for each structure. For DPO training, we use AdamW [26] as optimizer. We set $\beta$ to 0.5 for all experiments. We train our model on 8 * Nvidia A100 GPUs with a batch size of 128. Other parameters are summarized in Tab. 7.

## A.4 Preference Optimization of ProteinMPNN

To train the ProteinMPNN model, we slightly changed our dataset curation process and constructed a simpler task. Specifically, we set a new threshold $t_r = 0.8$ to control the process of classifying samples into chosen and rejected. Instead of training on all predictions like InstructPLM, we only trained ProteinMPNN on the predictions with a TM-Score lower than $t_r$. Meanwhile, the wild-type sequence has also been introduced as a chosen sample during training. As Fig. 5 shows, when trained on the original task ( 3) the model failed to converge, whereas the model successfully trained on a simpler task.

# B Additional Results

## B.1 Compare with Other Baselines

We provide a more detailed comparison with other baselines in Tab. 6.

## B.2 Evaluation using AlphaFold 3

To examine the robustness of our method across different structure predictors, we evaluated designs generated by our baseline and fine-tuned models using AlphaFold 3 [1]. Specifically, for each test structure where multi-round DPO was applied, we folded 10 designed sequences using AlphaFold 3, resulting in 200 sequences in total (100 from the baseline and 100 from the fine-tuned model). We report the average TM-Score and RMSD per structure, along with standard deviations, in the table 3 These results confirm that the improvements achieved by our method are consistent across

different folding models, suggesting that our approach captures structure-relevant features beyond model-specific feedback.

## B.3 Performance on CATH 4.3

To further assess the robustness of our method, we additionally evaluate it on selected structures from the newer CATH 4.3 release (Table 4). Specifically, we selected protein domains from newly added Classes in CATH 4.3 that are not present in CATH 4.2, ensuring no overlap with the training data not just at the Fold level, but at the broader Class level (the Fold and Class level of protein structure classification can be find in [32]). After filtering out sequences longer than 512 residues, we obtained a test set comprising 750 diverse and challenging structures. Compared to the original CATH 4.2 test set, which differs from training data at the Topology (Fold) level, this new dataset represents a more stringent test of generalization. The results show that our fine-tuned model consistently outperforms the baseline across all metrics. This performance gap is especially notable given the increased structural diversity and novelty in the CATH 4.3 set.

## B.4 Sequence Alignment of Visualized Samples

We performed sequence alignment (Fig. 7 to 10) of visualized samples in Fig. 2, illustrating each residue as a colored box to highlight matches (green), mismatches, and uncertain positions (gray). The results show that our fine-tuning process prioritizes the generation of sequences that fold into more accurate structures, even if it means deviating from the native sequence (low recovery rate).

## B.5 Improvements on different fold types.

We conducted a detailed analysis where we grouped the test structures in CATH 4.2 according to their Class annotations (**Mainly Alpha, Mainly Beta, Alpha-Beta, and Few Secondary Structures**). The average TM-scores before and after DPO fine-tuning are summarized in Table 5. As shown, DPO fine-tuning brings consistent improvements across all fold types, with especially notable gains in the Mainly Alpha and Mainly Beta categories. This demonstrates that our method not only improves structural quality overall but also robustly across a diverse range of protein topologies.

## B.6 More Results on Iterative Design.

For additional clarity, we monitor the running TM-Scores and recoveries of all 10 structures during training in Fig. 6. The plot demonstrates a clear upward trend, with substantial gains in the early rounds, followed by a gradual saturation effect as the model converges towards higher structure similarity. Concurrently, the recovery rate remains at a relatively low level, further emphasizing our method's ability to prioritize structural fidelity over sequence similarity, leading to the discovery of alternative sequences that fold more accurately.

Table 3: Performance on the CATH 4.2 test split.

| Model | ESMFold | | AlphaFold3 | |
| --- | --- | --- | --- | --- |
| | TM-Score | RMSD | TM-Score | RMSD |
| InstructPLM | 0.40 | 3.42 | 0.43 | 3.09 |
| InstructPLM-DPO | 0.70 | 2.25 | 0.68 | 2.29 |

Table 4: Performance on newly introduced Classes of CATH 4.3 dataset with respect to CATH 4.2.

| Model | Recovery | TM-Score | RMSD |
| --- | --- | --- | --- |
| InstructPLM | 41.85 | 0.57 | 2.89 |
| InstructPLM-DPO | 43.62 | 0.60 | 2.79 |

Table 5: TM-Score comparison across different fold types.

| Model | Mainly Alpha | Mainly Beta | Alpha Beta | Few Secondary Structures |
| --- | --- | --- | --- | --- |
| InstructPLM | 0.61 | 0.76 | 0.84 | 0.63 |
| InstructPLM-DPO | 0.66 | 0.80 | 0.86 | 0.65 |

Table 6: **Sequence design performance on CATH 4.2 test set.** We report the perplexity and recovery rate on the CATH 4.2 test set. The performance of our methods is shown in **bold**, while baselines without fine-tuning are indicated with an underline. We copied the results from [15] and reproduced the results of InstructPLM and ProteinMPNN in a unified codebase.

| Model | Perplexity on CATH 4.2↓ | | | Recovery Rate on CATH 4.2↑ | | |
|---|---|---|---|---|---|---|
| | Short | Single-chain | All | Short | Single-chain | All |
| StructGNN [19] | 8.29 | 8.74 | 6.40 | 29.44 | 28.26 | 35.91 |
| GraghTrans [19] | 8.39 | 8.83 | 6.63 | 28.14 | 28.46 | 35.82 |
| GCA [45] | 7.09 | 7.49 | 6.05 | 32.62 | 31.10 | 37.64 |
| GVP [20] | 7.23 | 7.84 | 5.36 | 30.60 | 28.95 | 39.47 |
| AlphaDesign [14] | 7.32 | 7.63 | 6.30 | 34.16 | 32.66 | 41.31 |
| ProteinMPNN [10] | 8.82 | 9.17 | 5.41 | 28.83 | 27.20 | 40.27 |
| ESM-IF [17] | 6.93 | 6.65 | 3.96 | 35.28 | 33.78 | 48.95 |
| PiFold [15] | 6.04 | 6.31 | 4.55 | 39.84 | 38.53 | 51.66 |
| LM-Design [53] | 6.77 | 6.46 | 4.52 | 37.88 | 42.47 | 55.65 |
| InstructPLM [36] | 3.22 | 3.17 | 2.63 | 40.88 | 40.87 | 53.58 |
| KW-Design [13] | 5.48 | 5.16 | 3.46 | 44.66 | 45.45 | 60.77 |
| **ProteinMPNN-DPO** | **8.40** | **8.50** | **5.09** | **29.19** | **27.41** | **41.29** |
| **InstructPLM-DPO** | **3.54** | **3.43** | **2.71** | **43.34** | **43.44** | **55.21** |

| Name | Single-round | Multi-round |
|---|---|---|
| Temperature | 1 | 1.1 |
| Top p | 0.9 | 1.0 |
| N | 20 | 200 |
| Number of rounds | 1 | 20 |
| DPO $\beta$ | 0.5 | 0.5 |
| Learning rate | 1e-5 | 1e-5 |
| Optimizer | AdamW | AdamW |
| Adam $\beta_1$ | 0.9 | 0.9 |
| Adam $\beta_2$ | 0.999 | 0.999 |
| Adam $\epsilon$ | 1e-8 | 1e-8 |
| Batch size | 128 | 128 |
| LoRA $r$ | 16 | 16 |
| LoRA $\alpha$ | 16 | 16 |
| Training steps | 4000 | 200 |

Table 7: Training configs

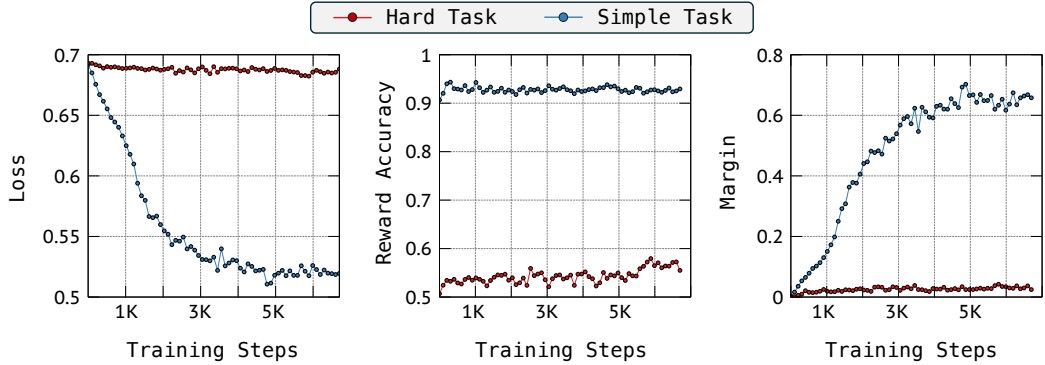

Figure 5: Metrics of ProteinMPNN training process.

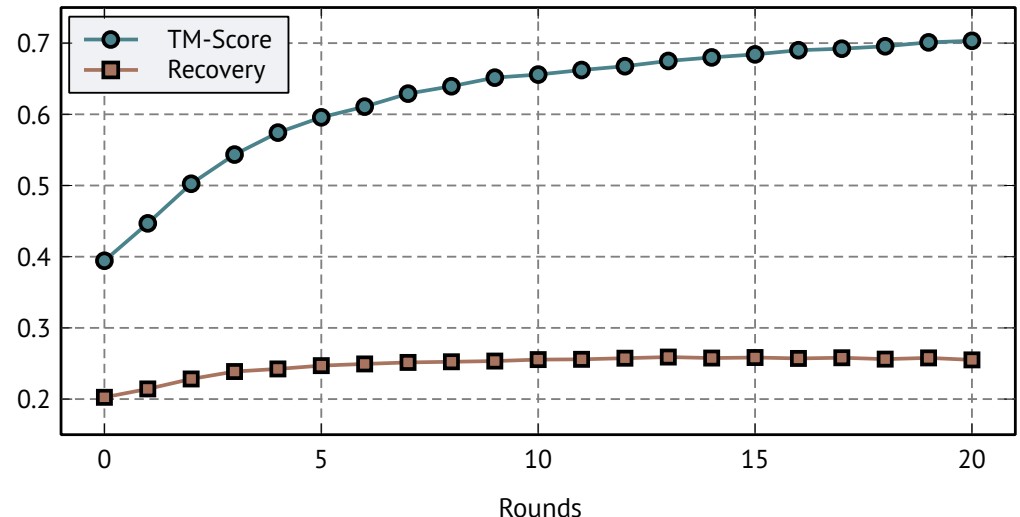

Figure 6: TM-Score and recovery rate changes across each round of iterative refinement.

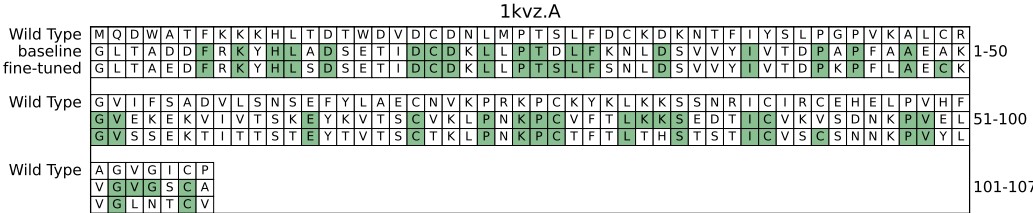

Figure 7: Sequence alignment of 1kvz.A. Baseline recovery rate: 0.36, TM-Score: 0.51. Finetuned recovery rate: 0.35, TM-Score 0.87.

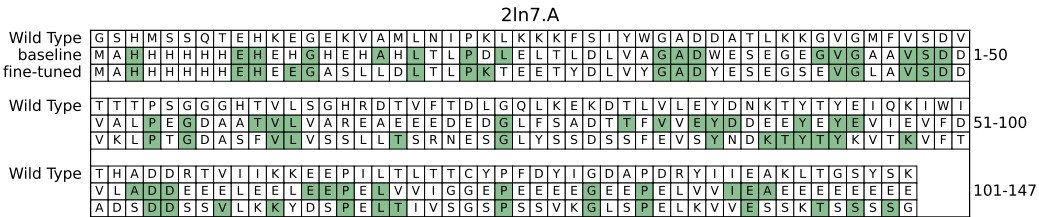

Figure 8: Sequence alignment of 2ln7.A. Baseline recovery rate: 0.30, TM-Score: 0.36. Finetuned recovery rate: 0.29, TM-Score 0.77.

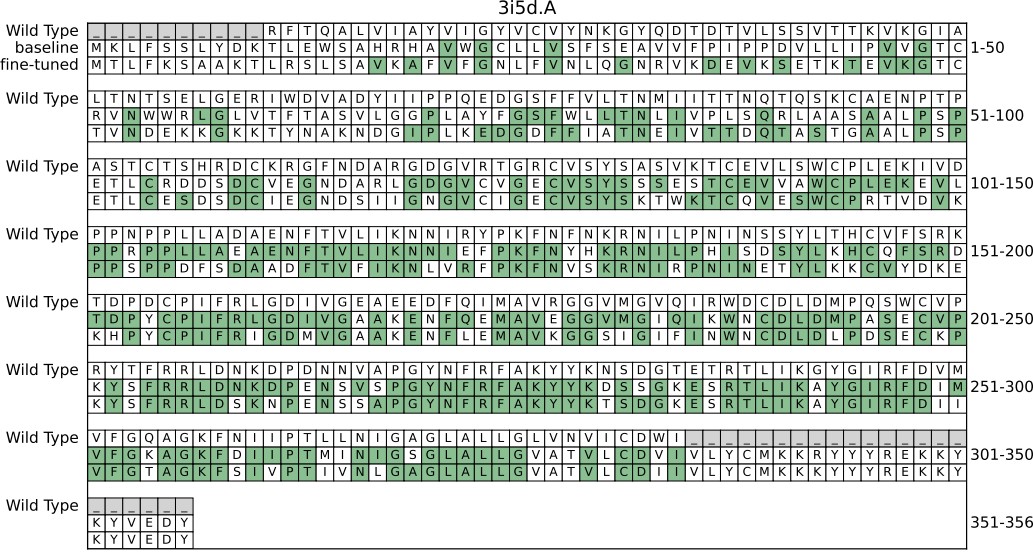

Figure 9: Sequence alignment of 3i5d.A. Baseline recovery rate: 0.57, TM-Score: 0.55. Finetuned recovery rate: 0.54, TM-Score 0.90.

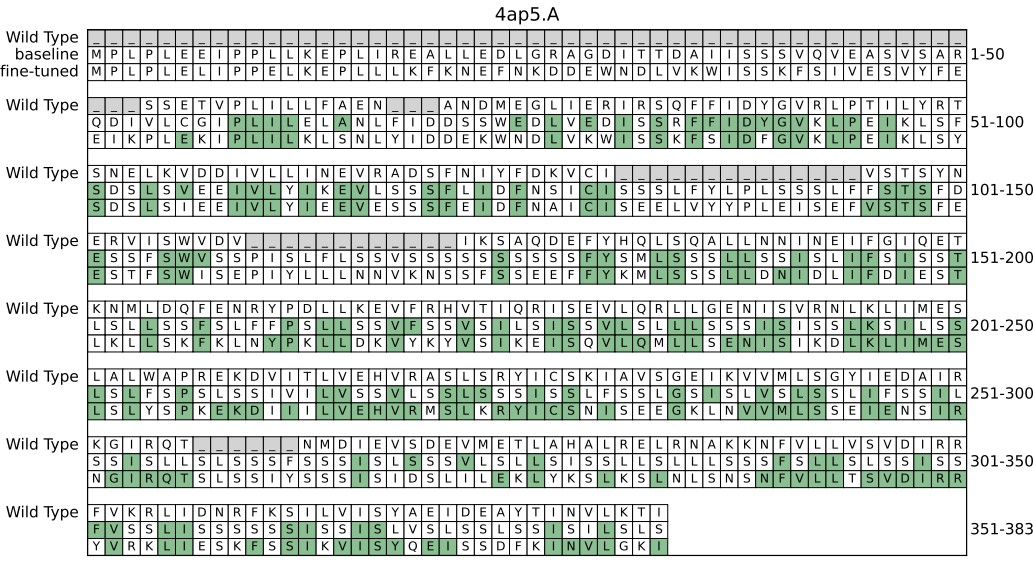

Figure 10: Sequence alignment of 4ap5.A. Baseline recovery rate: 0.38, TM-Score: 0.37. Finetuned recovery rate: 0.49, TM-Score: 0.94.

