# OpenReview forum: "Protein Inverse Folding From Structure Feedback"
_NeurIPS.cc/2025/Conference — NeurIPS 2025 poster_

### Official Review · Reviewer_k5nB · 2025-06-22

**Clarity:** 2
**Significance:** 3
**Originality:** 2
**Rating:** 4
**Confidence:** 2

**Summary:**

This paper introduces a novel framework for optimizing protein inverse folding models using Direct Preference Optimization (DPO) guided by structural feedback from a folding model. Given a target structure, the model samples multiple candidate sequences, predicts their folded structures, and scores them via TM-Align. Based on the TM-Score, sequences are classified into "chosen" and "rejected" and used to fine-tune the inverse folding model under a DPO objective. Experiments are done on the CATH 4.2, TS50, and TS500 datasets demonstrate improvements in both sequence recovery and TM-Score. The authors further investigate the effect of training scale, the number of contrastive samples, and iterative refinement via multi-round DPO.

**Questions:**

1. Figure 4 shows higher median TM-Scores for the fine-tuned model, but the heavy lower-tail suggests potential instability or skewness. Could the authors report statistical significance (e.g., Student's t-test), add distribution visualizations (e.g., violin/KDE plots), and clarify whether improvements are consistent across targets or driven by a few outliers? This would strengthen the claim of robust and meaningful improvement.

2. Why did you choose CATH 4.2 instead of the latest CATH 4.3? Would the model's performance be consistent if evaluated on the newer dataset? Please clarify if CATH 4.3 was excluded for compatibility, coverage, or reproducibility reasons.

3. Can you clarify the novelty of your method over prior DPO work? Are you the first to incorporate structural feedback into DPO for protein design? If not, how does your framework advance beyond existing works in terms of scale, architecture, or applicability?

**Ethical Concerns:**

["NO or VERY MINOR ethics concerns only"]

**Final Justification:**

The rebuttal addressed several of my main concerns. Statistical testing and finer-grained sampling analyses improved the rigor of the results, while the additional CATH 4.3 evaluation strengthened claims of generalization. Clarifications on novelty over prior DPO work also made the contribution clearer. Some issues remain—such as per-target consistency, biological relevance of TM-score optimization, and risk handling in multi-round DPO—but overall the paper is now better supported. I am raising my score to 4: Weak accept.

**Limitations:**

Yes. There is a paragraph titled “Limitations”.

**Quality:**

2

**Strengths And Weaknesses:**

*Strength*

-The authors evaluate their approach across multiple datasets (CATH 4.2, TS50, TS500), model architectures (ProteinMPNN, InstructPLM), and settings (single-round vs. multi-round, λ ablation), providing empirical evidence for robustness.

-The pipeline is entirely in silico and can be extended iteratively, making it well-suited for real-world protein design where experimental structures may be limited or unavailable.

-The use of equations, diagrams, and concrete implementation details make the method easy to follow and potentially reproducible.


*Weakness*

1. Several claims of "significant" improvement (e.g., Lines 251, 295) are made without statistical testing. In Figure 4, no significance tests are reported to validate the robustness of observed gains.
2. The evaluation of contrastive sample sizes in Figure 3 is based on only four discrete points (0, 20, 40, 80), which obscures potential non-linear effects or turning points in TM-Score and RMSD. This makes it difficult to draw reliable conclusions about how contrastive sample quantity affects structural fidelity.
3. TM-Score is used as the sole optimization and evaluation metric, yet structural similarity does not always correlate with functional correctness. Without assessing alternative measures (e.g., energy, function, or diversity), there is a risk of optimizing toward biologically irrelevant sequences.
4. Unaddressed Risks in Multi-Round DPO Training
 The iterative nature of multi-round DPO could amplify early-stage model biases or propagate errors. However, the paper does not discuss any safeguards such as validation-based early stopping, diversity checks, or convergence criteria to mitigate this risk.
5. While related work sections mention prior DPO-based efforts in enzyme and peptide design, the manuscript lacks a clear articulation of what makes structure-feedback-based protein design novel and technically distinct. The gap between this work and existing methods remains underexplored.
6. Minor presentation issues:
   - Table 1 lacks clarity on whether values are percentages.
   - Formatting choices (bold for fine-tuned models, underline for baselines) deviate from conventions and may confuse readers.
   - Line 169 uses "ESMfold" instead of the correct “ESMFold.”

---

> ### Author Rebuttal · Authors · 2025-07-28
>
> We sincerely thank the reviewer for the detailed and constructive feedback. We greatly appreciate the recognition of the strengths of our work, including its robustness, extensibility, and clarity of presentation. At the same time, we acknowledge the concerns raised regarding certain methodological and evaluative aspects. Below, we provide point-by-point responses to the weaknesses and questions.
>
> **Weakness 1, Question 1:** We thank the reviewer for the valuable suggestion regarding statistical rigor and visualization. As the reviewer noted, **Fig. 4** is presented as a **boxplot**, which includes the 25th percentile, median, and 75th percentile, providing a summary of the distribution and capturing potential skewness and variability in performance. To further address the reviewer’s concern, we perform t-test on the original data of Fig. 4:
>
> | t-stat | P-value  |
> | ------ | -------- |
> | -57.25 | < 1e-100 |
>
> We also have conducted an additional evaluation using **AlphaFold 3** to assess the robustness of our improvements across different structure predictors. Specifically, for each test structure where multi-round DPO was applied, we folded 10 designed sequences using AlphaFold 3, resulting in 200 sequences in total (100 from the baseline and 100 from the fine-tuned model). We report the average TM-Score and RMSD per structure, along with standard deviations, in the table below:
>
> |            | ESMFold-TMScore &#8593; | ESMFold-RMSD &#8595; | AlphaFold3-TMScore &#8593; | AlphaFold3-RMSD &#8595; |
> | ---------- | --------------- | ------------ | ------------------ | --------------- |
> | Baseline   | 0.40 (0.09)     | 3.42 (0.69)  | 0.43 (0.12)        | 3.09 (0.73)     |
> | Fine-tuned | 0.70 (0.18)     | 2.25 (0.89)  | 0.68 (0.17)        | 2.29 (0.83)     |
>
> We believe these results clearly reflect the significant performance improvements of our methods.
>
> **Weakness 2:** We conducted additional experiments with smaller increments in the number of contrastive samples (0, 10, 20, 30, 40, 80). The results show that performance improves up to around 20 contrastive samples, after which it begins to plateau or slightly degrade. Specifically, both TM-score and RMSD indicate that using 20 samples yields the best balance between structural accuracy and stability, confirming the trend observed in our original submission. These findings support our original choice and reinforce the importance of moderate contrastive sampling in DPO training.
>
> | Steps    | 0     | 10    | 20    | 30    | 40    | 80    |
> | -------- | ----- | ----- | ----- | ----- | ----- | ----- |
> | Recovery &#8593; | 53.48 | 55.09 | 55.21 | 55.04 | 55.15 | 55.18 |
> | TM-Score &#8593; | 0.78  | 0.80  | 0.81  | 0.81  | 0.78  | 0.78  |
> | RMSD &#8595;    | 2.11  | 1.98  | 1.97  | 1.98  | 2.08  | 2.08  |
>
> **Weakness 3:** We appreciate the reviewer’s concern regarding the biological relevance of the optimization target. We fully agree that incorporating functional and energetic criteria is crucial for real-world protein design. However, our primary goal in this work is to demonstrate that protein design algorithms can be effectively optimized *entirely in silico*, without relying on wild-type sequences or wet-lab supervision. To do so, we intentionally focus on TM-score as a well-established structural similarity metric, which provides a consistent and objective signal that can be computed automatically at scale. We view our approach as a proof of concept showing that feedback from structure prediction models alone can significantly improve inverse folding performance. We hope that our work can inspire future research to extend this framework by incorporating more biologically grounded objectives and stimulating the development of more accurate biological tools in the community.
>
> **Weakness 4:** We thank the reviewer for raising this important concern about potential risk amplification in multi-round DPO training. Indeed, recent work such as Ren et al. (2024) [1] has highlighted how iterative optimization can reinforce early-stage biases or drift from desired distributions. We will explicitly discuss this risk in the Limitations section of our revised manuscript. In our experiments, we report TM-Scores and Recovery throughout the multi-round training process to ensure that the model continues to learn meaningful structural features; please refer to **Fig. 6** for details. However, we would also like to emphasize that the primary contribution of our work is to demonstrate that fully in silico fine-tuning of protein language models using no external annotations or experimental data is feasible and can lead to significant performance gains. Our focus is thus on exploring and validating the possibility of DPO in protein design, rather than fully solving the challenge of safe or robust reinforcement-based learning in this domain. We leave the exploration of more robust and effective training frameworks for future work.
>
> **Weakness 5, Question 3:** We thank the reviewer for the thoughtful question regarding the novelty of our method compared to existing DPO-based approaches in protein design. First, we would like to emphasize that **all our results are obtained solely based on the model’s own predictions**, without relying on any external supervision such as wild-type sequences, evolutionary constraints, or wet-lab annotations. This contrasts with existing methods such as ProteinDPO[2] and ProtGPT2 (DPO)[3], which leverage annotated or curated datasets during training. Our framework instead demonstrates that protein language models can be improved entirely in silico by leveraging structure predictors as feedback, making the fine-tuning pipeline significantly more scalable and self-sufficient. Second, while there are a few concurrent works that also incorporate predicted structural feedback (e.g., using ESMFold) into the DPO process, these studies are typically limited in scope and scale. For instance, Stocco et al. focus only on a single enzyme [4], and Park et al. apply DPO to peptide sequences[5]. In contrast, our work performs DPO on over 1.8M predicted sequences and evaluates the fine-tuned model across diverse structural folds, demonstrating generalizable improvement beyond a narrow domain or sequence type. Third, our study also provides initial insights and foundations for designing larger-scale, fully in silico protein design frameworks by studying the scaling behavior of DPO. We hope this work can serve as a stepping stone toward more robust and autonomous pipelines in future research.
>
> We will further clarify these points in the revised version and highlight them in the Introduction and Related Work sections.
>
> **Weakness 6:** We thank the reviewer for pointing out these presentation issues; we will clarify in the revised version. However, in terms of the Table. 1, since our framework is model-agnostic, we aimed to emphasize the performance gains achieved by our fine-tuned models relative to their respective baselines, rather than compare across architectures.
>
> **Question 2:** We chose CATH 4.2 primarily for two reasons. First, our baseline models were trained on CATH 4.2, and to avoid any potential data leakage, we did not include CATH 4.3 structures in the training set. Using the same version ensures a fair and controlled comparison. Second, to better evaluate the generalization ability of our method, we conducted experiments on an out-of-distribution (OOD) set constructed from the CATH 4.3 release. Specifically, we selected protein domains from newly added **Classes** in CATH 4.3 that are not present in CATH 4.2, ensuring no overlap with the training data *not* just at the **Fold** level, but at the broader **Class** level. After filtering out sequences longer than 512 residues, we obtained a test set comprising 750 diverse and challenging structures. Compared to the original CATH 4.2 test set, which differs from training data at the Topology (Fold) level, this new dataset represents a more stringent test of generalization. The results show that our fine-tuned model consistently outperforms the baseline across all metrics. This performance gap is especially notable given the increased structural diversity and novelty in the CATH 4.3 set.
>
> | CATH 4.3 new Classes | Recovery &#8593; | TM-Score &#8593; | RMSD &#8595; |
> | -------------------- | -------- | -------- | ---- |
> | Baseline             | 41.85    | 0.57     | 2.89 |
> | Fine-tuned           | 43.62    | 0.60     | 2.79 |
>
> [1]: Ren, Yi, et al. "Bias amplification in language model evolution: An iterated learning perspective." *Advances in Neural Information Processing Systems* 37 (2024): 38629-38664.
>
> [2]: Widatalla, Talal, Rafael Rafailov, and Brian Hie. "Aligning protein generative models with experimental fitness via direct preference optimization." bioRxiv (2024): 2024-05.
>
> [3]: Mistani, Pouria, and Venkatesh Mysore. "Preference optimization of protein language models as a multi-objective binder design paradigm." arXiv preprint arXiv:2403.04187 (2024).
>
> [4]: Stocco, Filippo, et al. "Guiding generative protein language models with reinforcement learning." arXiv preprint arXiv:2412.12979 (2024).
>
> [5]: Park, Ryan, et al. "Improving inverse folding for peptide design with diversity-regularized direct preference optimization." arXiv preprint arXiv:2410.19471 (2024).

---

> > ### Comment · Reviewer_k5nB · 2025-08-01
> > **Comment**
> >
> > I thank the authors for their comprehensive rebuttal and additional analyses. The inclusion of statistical significance testing, expanded contrastive sampling evaluation, and OOD validation on CATH 4.3 improve the strength and clarity of the work. While a few concerns (e.g., biological relevance and risk mitigation in iterative training) could still benefit from further discussion in the final version, the core contribution is now more convincingly supported. I am therefore raising my score to 4.

---

### Official Review · Reviewer_xi2h · 2025-06-23

**Clarity:** 4
**Significance:** 3
**Originality:** 3
**Rating:** 5
**Confidence:** 4

**Summary:**

The paper proposes to use Direct Preference Optimization to fine tune existing models that perform protein design. That is, given a target structure, the idea is to hypothesize sequences that could fold into that structure using a pretrained model (they use proteinMPNN and InstructPLM). These sequences are folded (in silico) and then an alignment score is used to fine-tune the model used to generate hypothetical sequences. The effects of this on recovery rate, perplexity and structural similarity are studied.

**Questions:**

1) How do the improvements look across types of folds (e.g.see R2)?

2) How was the data for Fig 3 generated? Is that a specific protein, or averages? In general, it appears that as RMSD drops the recovery rate increases– that is de novo exploration reduces.

3) What is the scale of computation involved? E.g. per DPO iteration of each model (for different numbers of samples).

**Ethical Concerns:**

["NO or VERY MINOR ethics concerns only"]

**Limitations:**

Yes, but the limitations are quite generic.

**Paper Formatting Concerns:**

none.

**Quality:**

3

**Strengths And Weaknesses:**

The main strength of the paper is the relatively simple application to existing models to achieve significant improvements in structural similarity. Indeed, as highlighted, recovery rates are less important when the task is to design de novo proteins. Yet, it is good to see that the method maintains relatively healthy recovery rates as well as perplexity, while improving structural similarity. However, is unclear what the associated computational cost is. It is suggested to include and discuss the following references:

[1] Improving Protein Sequence Design through Designability Preference Optimization (https://arxiv.org/html/2506.00297v1)

[2] PDBench: evaluating computational methods for protein-sequence design (https://academic.oup.com/bioinformatics/article/39/1/btad027/6986968)

[3] Fine-tuning protein language models boosts predictions across diverse tasks
(https://www.biorxiv.org/content/10.1101/2023.12.13.571462v1)

---

> ### Author Rebuttal · Authors · 2025-07-28
>
> We thank the reviewer for the thoughtful comments and for highlighting the strengths of our work. We agree that recovery rate is not the most critical metric for de novo protein design, and we appreciate the recognition that our method improves structural similarity while maintaining reasonable recovery and perplexity levels. We also appreciate the suggested references. These works are indeed highly relevant to our study:
>
> [1] presents a concurrent approach applying DPO fine-tuning to protein language models, which aligns closely with our motivation and methodology.
>
> [2] emphasizes evaluating PLMs beyond recovery rate, which resonates with our view on the limitations of recovery as a sole metric.
>
> [3] explores parameter-efficient fine-tuning of PLMs across diverse tasks.
>
> We will include and discuss these works in the revised manuscript to better situate our contributions within the context of recent advances. Below, we address the reviewer’s questions and suggestions in detail:
>
> **Question 1**: We appreciate the reviewer’s interest in understanding how our method performs across different fold types. To this end, we conducted an additional experiment where we grouped the test structures in CATH 4.2 according to their **Class** annotations (*Mainly Alpha*, *Mainly Beta*, *Alpha-Beta*, and *Few Secondary Structures*). The average TM-scores (mean and standard deviation) before and after DPO fine-tuning are summarized as follows:
>
> |            | Mainly Alpha &#8593; | Mainly Beta &#8593; | Alpha Beta &#8593; | Few Secondary Structures &#8593; |
> | ---------- | ------------ | ----------- | ----------- | ------------------------ |
> | Baseline   | 0.61 (0.08)  | 0.76 (0.05) | 0.84 (0.04) | 0.63 (0.05)              |
> | Fine-tuned | 0.66 (0.06)  | 0.80 (0.04) | 0.86 (0.02) | 0.65 (0.03)              |
>
> As shown, DPO fine-tuning brings consistent improvements across all fold types, with especially notable gains in the *Mainly Alpha* and *Mainly Beta* categories. This demonstrates that our method not only improves structural quality overall, but does so robustly across a diverse range of protein topologies.
>
> **Question 2:** All the results in Fig. 3 are obtained from the CATH 4.2 test set, which contains 1,120 test structures that do not share the same Fold with the training set. For each structure, we sampled 10 sequences using both the baseline and fine-tuned models, and then computed the mean value of each metric (RMSD, TM-score, and recovery rate.) across the 10 sequences. Finally, the plotted points in Fig. 3 represent the averages over all 1,120 structures.  We would also like to clarify that the primary purpose of Fig. 3 is to analyze the scaling behavior of DPO with respect to different training configurations, including the number of training steps and training pairs. Our experiments reveal an interesting trend: while moderate amounts of preference pairs improve structural quality, excessive numbers of training pairs can actually degrade performance. This suggests that DPO benefits from high-quality, but not necessarily large-scale, feedback data, and highlights the importance of balancing signal strength with data quantity in preference-based optimization. For design sequences with high structure similarity but with low sequence recovery rate, we refer the reviewer to check our results on multi-round DPO in **Fig. 6**, which shows our methods achieve a higher TM-Score (0.7) but maintain low recovery (0.25).
>
> **Question 3:** The majority of the computational cost in our pipeline is concentrated in the **data construction phase**, particularly during sequence generation and structure prediction (folding). In contrast, the actual DPO fine-tuning stage is relatively lightweight. In our setting, a complete DPO training run takes approximately 16 hours on a server with 8×A100 GPUs. To promote future research, we will also release our full training dataset upon publication, which includes over 1.8 million sequences and their predicted structures generated by ESMFold.

---

> > ### Comment · Reviewer_xi2h · 2025-08-05
> > **Thank you.**
> >
> > Thank you for clarifying. It would help to incorporate these answers into the revision.

---

### Official Review · Reviewer_tPSY · 2025-07-02

**Clarity:** 4
**Significance:** 3
**Originality:** 3
**Rating:** 4
**Confidence:** 4

**Summary:**

This work applies Direct Preference Optimization (DPO) to the inverse folding task, leveraging TM-score as a feedback signal to enhance the foldability of designed protein sequences. The authors conducted comprehensive experiments to validate the effectiveness of the DPO strategy. In addition, they provide insightful analyses of the model, including its scaling behavior, an illustrative case study, and the impact of iterative training.

**Questions:**

See the Strengths and Weaknesses.

**Ethical Concerns:**

["NO or VERY MINOR ethics concerns only"]

**Limitations:**

Yes

**Paper Formatting Concerns:**

No.

**Quality:**

3

**Strengths And Weaknesses:**

The paper is well-written, with a comprehensive introduction, clear preliminaries, and detailed methodology. The authors also conducted in-depth analyses to evaluate and interpret the model. However, there are several minor concerns regarding the experimental section:

1. Lines 149–152 provide only a brief description of the application of DPO to ProteinMPNN. It would be helpful if the authors could elaborate further on this part for better clarity.

2. In line 189, it is unclear whether the authors explored other values for the hyperparameter λ, such as 0.1 or 0.01. Including such experiments would strengthen the analysis.

3. In Section 4.5, the results are reported only on ten test samples. It is recommended that the authors report performance metrics on the entire test set to provide a more comprehensive evaluation.

4. Since the main contribution of the paper is the application of DPO to inverse folding, demonstrating its effectiveness on additional baseline models would further substantiate the generalizability and robustness of the proposed approach.

---

> ### Author Rebuttal · Authors · 2025-07-30
>
> We thank the reviewer for their thoughtful and encouraging feedback. We are glad to hear that the paper is found to be well-written and the analyses insightful. We also appreciate the constructive suggestions for improving the clarity and completeness of our experiments. Below, we address the reviewer’s comments and concerns in detail.
>
> **Weakness 1:** To train the ProteinMPNN model, we slightly changed our dataset curation process and constructed a simpler task. Specifically, we set a new threshold t_r = 0.8 to control the process of classifying samples into chosen and rejected. Instead of training on all predictions like InstructPLM, we only trained ProteinMPNN on the predictions with a TM-Score lower than t_r . Meanwhile, in order to increase the diversity of the chosen sequences, the wild-type sequence has also been introduced as a chosen sample during training. Details can be found in the **Appendix A.1.4.** This result highlights an important insight: applying DPO effectively may require more powerful and expressive sequence design models. ProteinMPNN, while efficient and widely used, may lack sufficient capacity to fully benefit from DPO-style fine-tuning. We hope our findings can inspire the development of stronger structure-conditioned design algorithms that are better suited for preference-based optimization techniques like DPO.
>
> **Weakness 2:** We thank the reviewer for pointing out the importance of hyperparameter sensitivity analysis. We have conducted additional experiments varying the λ value and summarize the results (TM-Scores) below:
>
> |          | Short &#8593; | Single-Chain &#8593; | All &#8593; | TM-Score < 0.5 &#8593; | TM-Score > 0.5 &#8593; |
> | -------- | ------------- | -------------------- | ----------- | ---------------------- | ---------------------- |
> | λ = 10   | 0.57          | 0.61                 | 0.80        | 0.43                   | 0.88                   |
> | λ = 1    | 0.58          | 0.63                 | 0.81        | 0.45                   | 0.89                   |
> | λ = 0.1  | 0.58          | 0.62                 | 0.81        | 0.45                   | 0.88                   |
> | λ = 0.01 | 0.57          | 0.62                 | 0.81        | 0.44                   | 0.88                   |
> | λ = 0    | 0.57          | 0.62                 | 0.81        | 0.44                   | 0.88                   |
>
> These results suggest that the model is relatively robust to λ values in a wide range. While λ = 1 yields slightly better performance overall, especially on challenging (TM-Score < 0.5) cases, smaller values like 0.1 or 0.01 also perform comparably well. We will include these findings in the revised manuscript to strengthen our analysis.
>
> **Weakness 3:** We thank the reviewer for this suggestion. In Section 4.5, we initially selected 10 samples to qualitatively examine whether multi-round DPO can help the model successfully design structures that it previously **failed** to recover, focusing on cases where improvement is most meaningful. These samples were chosen based on relatively low baseline performance to better illustrate the effect of iterative optimization. We agree that evaluating on a larger and randomly selected subset provides a more comprehensive view. To this end, we extended the multi-round DPO experiment to 100 randomly sampled proteins from the CATH 4.2 test set. The results are summarized below:
>
> |                  | Round 0 | Round 2 | Round 4 | Round 6 | Round 8 | Round 10 |
> | ---------------- | ------- | ------- | ------- | ------- | ------- | -------- |
> | TM-Score &#8593; | 0.72    | 0.75    | 0.77    | 0.78    | 0.77    | 0.77     |
> | RMSD &#8595;     | 2.56    | 2.39    | 2.23    | 2.19    | 2.21    | 2.23     |
>
> We observe consistent improvement in both TM-Score and RMSD during the early rounds, indicating that multi-round DPO can generalize its benefits beyond the small and hard examples. However, the performance appears to saturate in the later rounds. This indicates that multi-round DPO is particularly effective in challenging cases with low initial performance, while for easier or already well-predicted samples, the benefit plateaus more quickly. We will incorporate these results into the revised version to strengthen our empirical analysis.
>
> **Weakness 4:** We agree that evaluating additional baseline models would further demonstrate the generalizability of our approach. However, due to time and resource constraints during the rebuttal period, we were unable to extend our experiments to new baseline architectures. Notably, our current results already cover a diverse range of datasets and evaluation settings, including challenging OOD structures from CATH 4.3, which collectively demonstrate the robustness and structural awareness gained through DPO training. We consider this a promising direction for future work.

---

### Official Review · Reviewer_UQVk · 2025-07-02

**Clarity:** 3
**Significance:** 2
**Originality:** 2
**Rating:** 4
**Confidence:** 4

**Summary:**

This paper presents a method for improving protein inverse folding by leveraging feedback from a folding model using Direct Preference Optimization (DPO). Given a target structure, the inverse folding model generates candidate sequences, which are scored by a folding model based on structural similarity (TM-score). These scores are used to create preference pairs (chosen vs. rejected), which guide DPO fine-tuning of the inverse folding model. The approach improves both sequence recovery and structural accuracy across datasets (CATH 4.2, TS50, TS500), and shows strong gains in multi-round iterative refinement. The method is fully in silico, scalable, and applicable to both small and large protein models.

**Questions:**

Questions for the authors:

1. Have you analyzed how performance scales with the number of unique training structures (i.e., a full learning curve)?
2. How do you control for low-quality or ambiguous contrastive samples? Have you tried thresholding TM-Score or using clustering?
3. Does structural diversity in the training set affect generalization?
4. Have you tested sensitivity to the choice of folding model for feedback (e.g., ESMFold vs. AlphaFold)?

Suggestions:

1. Run a learning curve experiment varying the number of training structures to assess data efficiency and saturation.
2. Filter training pairs using absolute TM-Score thresholds instead of a 50/50 split to improve signal quality.
3. Combine structural feedback with additional objectives (e.g., stability, binding affinity) to enrich preference signals.
4. Evaluate generalization to unseen folds or novel structural classes to test robustness.
5. Visualize how DPO shifts sequence distributions (e.g., t-SNE of embeddings) to verify preference-driven learning dynamics.
6. The results in Figure 3 on scaling contrastive samples are insightful, but the granularity is limited. The jumps from 0 to 20 to 40+ samples are too coarse to reveal meaningful trends or tipping points. A more fine-grained analysis (e.g., increments of 5 or 10) would help clarify how performance evolves and where degradation begins.
7. Table 1 reports performance metrics without standard deviations, making it unclear whether improvements are statistically significant. To address this, multiple training runs with bootstrapped training set samples should be performed. Statistical significance can then be assessed using critical difference diagrams (e.g., via the `scikit-posthocs` package: [https://scikit-posthocs.readthedocs.io/en/latest/generated/scikit\_posthocs.critical\_difference\_diagram.html](https://scikit-posthocs.readthedocs.io/en/latest/generated/scikit_posthocs.critical_difference_diagram.html)). This would provide a robust comparison across models and datasets.

**Ethical Concerns:**

["NO or VERY MINOR ethics concerns only"]

**Final Justification:**

This paper presents a method for improving protein inverse folding by applying Direct Preference Optimization (DPO) guided by structure-based feedback from folding models. The rebuttal was thorough and addressed several of my concerns, strengthening the submission. However, some limitations remain, particularly in terms of evaluation depth and statistical rigor.

### Resolved Issues:

* **Learning curve and data efficiency:** The authors provided a new ablation showing graceful degradation with reduced structural diversity, demonstrating robustness to dataset size.
* **Generalization to novel folds:** The authors clarified their original split ensured fold-level disjointness and added new results on CATH 4.3, confirming performance gains on structurally novel targets.
* **Finer-grained sampling analysis (Fig. 3):** Additional experiments with intermediate numbers of contrastive samples clarified the performance plateau and supported their original design choices.
* **Robustness to folding model choice:** Evaluation using AlphaFold3 confirmed that gains are not specific to ESMFold, increasing confidence in the generality of the method.

### Remaining Issues:

* **Statistical significance and variability:** The authors reported per-target standard deviations at inference time but did not perform training-level replication or statistical testing (e.g., bootstrapping, critical difference diagrams). This limits confidence in the reported improvements, particularly in Table 1.
* **Lack of broader objective integration:** The framework remains narrowly focused on structural similarity. While extensibility was acknowledged, no preliminary results on alternative design objectives (e.g., stability or binding) were provided.
* **No sequence space analysis:** The suggestion to visualize how DPO shifts sequence distributions was acknowledged but deferred, missing an opportunity to clarify model behavior.

### Final Assessment:

This is a technically solid and clearly written paper that addresses a relevant problem with a practical, scalable method. The rebuttal meaningfully improved the submission, particularly in evaluating generalization and training dynamics. However, due to limited statistical rigor and a somewhat narrow scope of evaluation, I maintain a **Borderline Accept (4)** rating.

**Limitations:**

Yes

**Quality:**

2

**Strengths And Weaknesses:**

**Strengths**

**Quality:**

* Methodologically solid; applies Direct Preference Optimization (DPO) to inverse folding using feedback from folding models.
* Demonstrates consistent improvements in structure similarity (TM-score) and sequence recovery across models and datasets.
* Multi-round DPO refinement shows strong gains, particularly on challenging protein structures.
* Fully in silico pipeline avoids reliance on experimental data, supporting scalability.

**Clarity:**

* The paper is well organized and the method is clearly described.
* Figures are helpful, especially the pipeline diagram and TM-score visualizations.

**Significance:**

* Addresses a key gap: current inverse folding models over-optimize for sequence similarity.
* Shows that structure feedback can improve sequence design quality in practice.
* Results have practical implications for scalable, structure-aware protein design.

---

**Weaknesses**

**Quality:**

* Table 1 lacks standard deviations; no statistical testing is reported.

  * Authors should use bootstrapped training set sampling and include critical difference diagrams to establish significance.
* Figure 3 is too coarse (e.g., 0, 20, 40 samples); finer-grained evaluation would clarify where data scaling starts to hurt performance.
* No learning curve analysis with respect to the number of unique training structures; it's unclear whether performance is limited by data volume or model capacity.

**Clarity:**

* Some figures (e.g., Fig. 3) lack clarity in axis labels and legends.
* The role and tuning of λ in balancing DPO and SFT losses could be better explained.

**Significance:**

* Generalization to novel folds or unseen structural classes is not evaluated.
* No integration of functional objectives (e.g., stability, binding) beyond structural similarity.

**Originality:**

* The use of DPO in protein language models is not new.

  * Section 5 notes similar approaches (e.g., ProteinDPO, DPO\_pLM), though those focus on other objectives (e.g., stability, function) or smaller scales.
  * This work's main novelty is the specific application of **structure-based feedback** and fully synthetic training, which is incremental rather than fundamentally new.

---

> ### Author Rebuttal · Authors · 2025-07-29
>
> We thank the reviewer for their thorough and insightful comments. We appreciate the recognition of our method’s strengths in terms of clarity, methodological rigor, and the practical significance of integrating folding-based feedback via Direct Preference Optimization (DPO). Below we address each concern in detail.
>
> **Question 1, Suggestion 1, Question 3**: We thank the reviewer for raising the important question regarding the effect of structural diversity in the training set on generalization. To investigate this, we conducted an ablation study by subsampling the number of training structures from the full CATH 4.2 training set. Specifically, we randomly selected **50% (8,987 structures)** and **25% (4,515 structures)** of the full set, and re-ran the training pipeline under the same protocol, 0% indicates baseline performance before fine-tuning. The results on the held-out CATH 4.2 test set are summarized below:
>
> | Smaller Dataset  | 0%    | 25%   | 50%   | 100%  |
> | ---------------- | ----- | ----- | ----- | ----- |
> | Recovery &#8593; | 53.48 | 54.51 | 55.06 | 55.21 |
> | TM-Score &#8593; | 0.78  | 0.79  | 0.81  | 0.81  |
> | RMSD &#8595;     | 2.11  | 2.02  | 1.98  | 1.97  |
>
> These results indicate that our method remains robust even when trained on a reduced number of structures, for example, it achieves comparable performance when using only 50% of the training set. However, as the number of training structures is further reduced, a noticeable drop in performance is observed. We will include these findings in the revised version to better illustrate the relationship between training set diversity and model generalization.
>
> **Question 2, Suggestion 2:** We use a 50:50 split instead of thresholding for the following reasons. First, the performance of protein design models varies significantly across different target structures. Applying a fixed TM-score threshold, for example, 0.8, would result in many structures lacking valid negative responses, as the model may generate only sequences above the threshold. This would reduce structural diversity in training. Second, for structures where the model performs poorly, it becomes difficult to find valid chosen responses, which leads to severe class imbalance. In contrast, the 50:50 split can be seen as a dynamic, per-structure threshold that ensures a balanced set of preference pairs for each structure. We acknowledge that this approach may introduce some noisy signals—for instance, cases where chosen responses have relatively low TM-scores or rejected responses have high TM-scores. However, since DPO is a contrastive learning objective, it only requires that the chosen sample be *relatively* better than the rejected one, not absolutely good. Given these considerations, we favored a simple but robust strategy over more complex selection mechanisms.
>
> **Question 4:** We agree with the reviewer that AlphaFold generally provides more accurate structure predictions than ESMFold. However, due to its significantly higher computational cost, AlphaFold is not well-suited for integration into the training loop. To examine the robustness of our method across different structure predictors, we evaluated designs generated by our baseline and fine-tuned models using AlphaFold 3.  Specifically, for each test structure where multi-round DPO was applied, we folded 10 designed sequences using AlphaFold 3, resulting in 200 sequences in total (100 from the baseline and 100 from the fine-tuned model). We report the average TM-Score and RMSD per structure, along with standard deviations, in the table below:
>
> |            | ESMFold-TMScore &#8593; | ESMFold-RMSD &#8595; | AlphaFold3-TMScore  &#8593; | AlphaFold3-RMSD &#8595; |
> | ---------- | ----------------------- | -------------------- | --------------------------- | ----------------------- |
> | Baseline   | 0.40 (0.09)             | 3.42 (0.69)          | 0.43 (0.12)                 | 3.09 (0.73)             |
> | Fine-tuned | 0.70 (0.18)             | 2.25 (0.89)          | 0.68 (0.17)                 | 2.29 (0.83)             |
>
> These results confirm that the improvements achieved by our method are consistent across different folding models, suggesting that our approach captures structure-relevant features beyond model-specific feedback.
>
> **Suggestion 3:** Combining structural feedback with additional objectives such as stability or binding affinity is indeed a promising direction. One of the main goals of our work is to evaluate whether fully in silico simulation alone can improve protein design performance, without relying on wet-lab feedback. Importantly, our framework does not make strong assumptions about the scoring function and is compatible with a wide range of design objectives. In principle, predicted metrics such as stability or binding affinity can also be seamlessly incorporated into our training pipeline. However, such metrics often require costly simulations or are less mature compared to structure-based evaluation. We leave the integration and benchmarking of alternative objectives to future work, and we hope our results can inspire broader exploration of these directions in the community.
>
> **Suggestion 4:** The CATH 4.2 test set already contains folds that are disjoint from the training set, as we follow the data split protocol from [1], which ensures no overlap at the **CAT (Topology/Fold)** level [2]. Therefore, the results reported in Table 1 can be interpreted as evaluating generalization to unseen folds. To further assess the robustness of our method, we additionally evaluate it on selected structures from the newer CATH 4.3 release. Specifically, we selected protein domains from newly added **Classes** in CATH 4.3 that are not present in CATH 4.2, ensuring no overlap with the training data *not* just at the **Fold** level, but at the broader **Class** level. After filtering out sequences longer than 512 residues, we obtained a test set comprising 750 diverse and challenging structures. Compared to the original CATH 4.2 test set, which differs from training data at the Topology (Fold) level, this new dataset represents a more stringent test of generalization. The results show that our fine-tuned model consistently outperforms the baseline across all metrics. This performance gap is especially notable given the increased structural diversity and novelty in the CATH 4.3 set.
>
> | CATH 4.3 new Class | Recovery &#8593; | TM-Score &#8593; | RMSD &#8595; |
> | ------------------ | ---------------- | ---------------- | ------------ |
> | Baseline           | 41.85            | 0.57             | 2.89         |
> | Fine-tuned         | 43.62            | 0.60             | 2.79         |
>
> **Suggestion 5:** We appreciate the reviewer’s suggestion to visualize how DPO shifts the sequence distribution (e.g., using t-SNE on embeddings) to better understand the learning dynamics. Due to current constraints during the rebuttal phase, we are unable to include such analyses at this stage. However, we fully agree that improved visualizations would strengthen the work, and we plan to incorporate them in a future revision of the paper.
>
> **Suggestion 6:** We conducted additional experiments with smaller increments in the number of contrastive samples (0, 10, 20, 30, 40, 80). The results show that performance improves up to around 20 contrastive samples, after which it begins to plateau or slightly degrade. Specifically, both TM-score and RMSD indicate that using 20 samples yields the best balance between structural accuracy and stability, confirming the trend observed in our original submission. These findings support our original choice and reinforce the importance of moderate contrastive sampling in DPO training.
>
> | Steps            | 0     | 10    | 20    | 30    | 40    | 80    |
> | ---------------- | ----- | ----- | ----- | ----- | ----- | ----- |
> | Recovery &#8593; | 53.48 | 55.09 | 55.21 | 55.04 | 55.15 | 55.18 |
> | TM-Score &#8593; | 0.78  | 0.80  | 0.81  | 0.81  | 0.78  | 0.78  |
> | RMSD &#8595;     | 2.11  | 1.98  | 1.97  | 1.98  | 2.08  | 2.08  |
>
> **Suggestion 7:** We appreciate the reviewer’s suggestion regarding statistical significance. Due to computational constraints, we were not able to retrain the model multiple times with different seeds or bootstrapped training sets. However, the results reported in Table 1 are already averaged over 10 independently sampled sequences per target structure, providing a robust estimate of performance variability at the inference level. Below, we report the standard deviations across these 10 samples for each metric:
>
> |            | Recovery (mean, std) &#8593; | TM-Score (mean, std) &#8593; | RMSD (mean, std) &#8595; |
> | ---------- | ---------------------------- | ---------------------------- | ------------------------ |
> | Baseline   | 53.58, 0.03                  | 0.78, 0.05                   | 2.11, 0.35               |
> | Fine-tuned | 55.21, 0.02                  | 0.81, 0.03                   | 1.97, 0.28               |
>
> We believe these per-target statistics provide meaningful evidence of the improvements achieved by our method, even without full training-level replication. We agree that training-level variance is important and plan to include a more comprehensive statistical analysis in future work when resources permit.
>
> [1]: Ingraham, John, et al. "Generative models for graph-based protein design." *Advances in neural information processing systems* 32 (2019).
>
> [2]: Orengo, Christine A., et al. "CATH–a hierarchic classification of protein domain structures." *Structure* 5.8 (1997): 1093-1109.

---

### Note · Authors · 2025-08-12

We thank the reviewers for their constructive feedback. In the rebuttal, we have expanded our experiments in three main directions:

**Additional ablations:** We performed finer-grained analyses on key hyperparameters, providing a more detailed understanding of their influence on model performance.

**Generalization tests:** We evaluated the framework under multiple new settings: (i) evaluate fine-tuned (trained on ESMFold) models using AlphaFold 3, (ii) larger-scale multi-round training, and (iii) evaluation on more challenging data (training on CATH 4.2 and testing on CATH 4.3). These results confirm the robustness of our method under both distribution shifts and more challenging scenarios.

**Enhanced result analysis:** We examined performance improvements across different folds and reported standard deviations to give a clearer view of stability and variance.

Overall, these new results reinforce our main claim: the proposed framework can achieve measurable performance gains for protein design entirely under a fully in silico setting, without relying on wet-lab feedback. We believe this demonstrates both the practicality and scalability of our approach and hope it can serve as a useful reference for future large-scale framework design.

---

### Decision · Program_Chairs · 2025-09-17

**Decision:**

Accept (poster)

**Comment:**

This paper develops an improved protein inverse folding model using DPO guided by structural feedback from a folding model (TM-Score).

Strength
-The authors evaluate their approach across multiple datasets (CATH 4.2, TS50, TS500) and show strong  empirical evidence for robustness.
- Multi-round DPO refinement shows strong gains, particularly on challenging protein structures.
- Fully in silico pipeline avoids reliance on experimental data, supporting scalability.

Weakness
- Concern about the use of TM-Score as reward and evaluation metric.
- Potential bias introduced and amplified with the iterative training.
- The method itself is limited in technical novelty.